# Advancing Allogeneic Hematopoietic Stem Cell Transplantation Outcomes through Immunotherapy: A Comprehensive Review of Optimizing Non-CAR Donor T-Lymphocyte Infusion Strategies

**DOI:** 10.3390/biomedicines12081853

**Published:** 2024-08-14

**Authors:** Stefania Braidotti, Marilena Granzotto, Debora Curci, Barbara Faganel Kotnik, Natalia Maximova

**Affiliations:** 1Department of Pediatrics, Institute for Maternal and Child Health-IRCCS Burlo Garofolo, 34137 Trieste, Italy; stefania.braidotti@burlo.trieste.it; 2Azienda Sanitaria Universitaria Giuliano Isontina (ASU GI), 34125 Trieste, Italy; marilena.granzotto@asugi.sanita.fvg.it; 3Advanced Translational Diagnostic Laboratory, Institute for Maternal and Child Health-IRCCS Burlo Garofolo, 34137 Trieste, Italy; debora.curci@burlo.trieste.it; 4Department of Hematology and Oncology, University Children’s Hospital, 1000 Ljubljana, Slovenia; barbara.faganel@gmail.com

**Keywords:** hematopoietic stem cell transplantation, T lymphocyte, memory T cells, virus-specific T cells, graft-versus-host disease, graft-versus-leukemia

## Abstract

Optimized use of prophylactic or therapeutic donor lymphocyte infusions (DLI) is aimed at improving clinical outcomes in patients with malignant and non-malignant hematological diseases who have undergone allogeneic hematopoietic stem cell transplantation (allo-HSCT). Memory T-lymphocytes (CD45RA−/CD45RO+) play a crucial role in immune reconstitution post-HSCT. The infusion of memory T cells is proven to be safe and effective in improving outcomes due to the enhanced reconstitution of immunity and increased protection against viremia, without exacerbating graft-versus-host disease (GVHD) risks. Studies indicate their persistence and efficacy in combating viral pathogens, suggesting a viable therapeutic avenue for patients. Conversely, using virus-specific T cells for viremia control presents challenges, such as regulatory hurdles, cost, and production time compared to CD45RA-memory T lymphocytes. Additionally, the modulation of regulatory T cells (Tregs) for therapeutic use has become an important area of investigation in GVHD, playing a pivotal role in immune tolerance modulation, potentially mitigating GVHD and reducing pharmacological immunosuppression requirements. Finally, donor T cell-mediated graft-versus-leukemia immune responses hold promise in curbing relapse rates post-HSCT, providing a multifaceted approach to therapeutic intervention in high-risk disease scenarios. This comprehensive review underscores the multifaceted roles of T lymphocytes in HSCT outcomes and identifies avenues for further research and clinical application.

## 1. Allogeneic Hematopoietic Stem Cell Transplantation Landscape

Allogeneic hematopoietic stem cell transplantation (allo-HSCT) is a highly effective life-saving treatment for hematological disorders and malignancies to restore normal hematopoiesis and treat malignancies in pediatrics and adults. It offers a potential cure and long-term disease control for leukemia and inherited bone marrow failures [1,2,3]. This therapeutic approach involves the infusion of stem cells from a healthy donor, preferably identical or “matched” for human leukocyte antigen (HLA), to reconstitute hematopoiesis in patients with dysfunctional bone marrow. Only 25% of patients who need an allograft have an HLA-identical sibling available as a donor. However, alternative stem cell sources for patients lacking a fully matched sibling donor include a matched unrelated donor, a partially matched cord blood unit, or a family member sharing one HLA haplotype with the recipient [4,5]. Differences in HLA classes between donors and recipients can lead to reactions against mismatched HLA molecules, resulting in serious issues compromising the success of the transplant procedure, such as graft failure and acute or chronic graft-versus-host disease (GVHD) incidence [6]. 

The bone marrow microenvironment is crucial for the success of allo-HSCT. Both hematopoietic and immune microenvironments have been increasingly recognized as being significant in improving the safety of HSCT and overcoming post-transplant complications [7,8]. The post-transplant immune regulatory microenvironment is highly complex since it involves a dynamic interplay between various immune cell types, including T lymphocytes, natural killer (NK) cells, dendritic cells, macrophages, and regulatory T cells (Tregs). The interactions among these cells play a crucial role in reconstituting the immune system, preventing infections, managing GVHD, and achieving a graft-versus-leukemia (GVL) effect. During transplantation, even mesenchymal stem/stromal cells (MSCs) help to expedite hematopoietic reconstitution by secreting chemokines that induce hematopoietic stem cells to nest across blood vessels and into bone marrow; moreover, MSCs could regulate T cell activation and reduce NK cell ratio to alleviate GVHD [9,10]. T lymphocytes interact with various immune cells through antigen presentation, cytokine signaling, and direct cell-cell contact to coordinate the immune response following stem cell transplantation. The balance between pro-inflammatory and anti-inflammatory responses is carefully regulated through cytokine networks, regulatory cell functions, checkpoint inhibitors, and feedback mechanisms. This regulation ensures effective defense against pathogens while preventing excessive inflammation and autoimmunity [11,12,13]. Moreover, T lymphocytes have a role in cytokine production (e.g., IL-2. IL-4. IL-5, IL-13, IFN-γ, TNFα) that influence the activity of other immune cells. For instance, T helper lymphocytes secrete interleukins that stimulate B cells, macrophages, and other T lymphocytes. Furthermore, Tregs maintain immune homeostasis by suppressing the activity of other immune cells, preventing excessive immune responses through direct cell-cell contact, secretion of inhibitory cytokines (e.g., IL-10, TGF-β), and the modulation of dendritic cell function [7,14,15]. 

In the context of allogeneic HSCT, T lymphocytes (CD3+) play a dual role. They are considered contaminants because they can cause potentially lethal GVHD, but they also serve as active components in controlling residual blasts (GVL) and opportunistic and viral infections. Peripheral blood stem cell product has 1-log more T cell dose than bone marrow and has indeed been associated with a higher risk of GVHD [16,17]. The tolerable dose of T cells varies widely, and HLA-matching is the most important predictive factor. Therefore, several approaches have been developed to manage T cell depletion (in vivo and ex vivo) and to contain immunological alloreactivity by immunosuppressive drugs.

Notably, the additional use of pre-emptive, prophylactic, or therapeutic donor lymphocyte infusions (DLI) in the post-transplant setting has been pivotal. This practice has evolved significantly over the years. Unmanipulated DLI is proven after both myeloablative and non-myeloablative HSCT to treat and prevent primary disease relapse, to establish full donor chimerism, and to treat and prevent infections, speeding up recipients’ immuno-recovery [18,19]. These infusions, which contain a heterogeneous mix of immune cells from the donor, were straightforward to prepare and administer. However, their use came with a considerable risk of GVHD, which posed a significant challenge in post-transplant care. Advances in medical technology have transformed this landscape, enabling the development of clinical-grade manipulated DLIs using closed, automated systems compliant with Good Manufacturing Practice (GMP) guidelines. Sophisticated cell sorting processing is made possible with CliniMACS^®^ Plus (semi-automated) or CliniMACS^®^ Prodigy (fully automated).

In the following paragraphs, we will carefully analyze these aspects, providing detailed insights into the critical role played by donor T lymphocytes regarding specific subtypes. We will also explore the wide range of impactful potential challenges and opportunities in the post-transplant setting.

## 2. Role of Manipulated Donor-T Lymphocytes to Enhance Immune Reconstitution

The timely reconstitution and recovery of the donor-derived immune system function is crucial for the long-term survival of patients undergoing allo-HSCT [20,21]. However, it depends on several factors, such as the conditioning regimen, source of stem cells, total nucleated cell dose and CD34+ cell dose within the graft source, transplant composition, and immunosuppression approaches [22,23]. Generally, innate immunity recovers rapidly within 20–30 days after allo-HSCT, with monocytes first, followed by granulocytes and NK cells within 100 days. In contrast, adaptive immunity, which consists of cellular and humoral immunity, takes 1–2 years to recover fully [24]. In this context, DLI can accelerate the recovery of the immune system by providing a more robust and functional cell population, which is essential to reduce the period of immunodeficiency and vulnerability to post-transplant infections. Memory T cells are particularly valuable for their rapid response to previously encountered antigens and their ability to provide long-term immunity [25]. 

Clinical-grade memory T cells (CD45RA−/CD45RO+) can be successfully obtained thanks to manufacturing platforms, such as CliniMACS^®^ Plus or CliniMACS^®^ Prodigy. The depletion of CD45RA+ cells begins with the collection of the starting material, which consists of leukapheresis product or whole blood, collected from the donor. The collected cells are then processed using clinical-grade magnetic CD45RA beads to remove CD45RA-expressing T cells selectively [26]. The CD45RA+ depletion results in a greater than 4-log reduction in the number of target cells and passively enrichment for memory T cells ready to use [27]. 

The infusion of memory T cells is proven to be safe and effective. Triplett et al. demonstrated that infusions of donor memory T cells (CD45RA−/CD45RO+) after allo-HSCT leads to a significant increase in neutrophil engraftment, quick conversion to full donor chimerism, persistence, and prominent expansion of selected T cell clones, with robust reconstitution of innate and memory immunity, as well as enhanced protection against viremia [28]. Shook et al. also observed that transplantation of haploidentical stem cells combined with CD45RA-negative lymphocytes provides rapid engraftment and excellent tolerability in the pediatric cohort analyzed [29]. Other authors have also observed this in hematological malignant and non-malignant diseases. Beyond the transplantation setting, the recommended number of cells to be infused has not been defined by international guidelines, nor has there been any consensus about the precise time when infusion is most appropriate. However, some examples have proven to be safe. Several clinical studies have been published regarding the passive enrichment of memory T cells and their infusion after HSCT in pediatrics and adults in a highly variable range of memory T cells/kg through different timeframes (Table 1). Overall, the data highlights a reduced risk of GVHD and improved outcomes due to the enhanced reconstitution of innate immunity and memory and increased protection against viremia and Aspergillosis [30].

**Table 1 biomedicines-12-01853-t001:** Role of memory donor T lymphocytes to enhance immune reconstitution and protect against infection. aGVHD: acute graft-versus-host disease; cGVHD: chronic graft-versus-host disease; CMV: cytomegalovirus; EFS: event-free survival; MUD: matched unrelated donor; NRM: non-relapse mortality; NK, natural killer; OS: overall survival; PFS: progression-free survival.

Patients/ Primary Disease/Age at HSCT *Median (Range), Years*	HSCT Protocol	CD34+ Cell Dose *Median (Range)*	Memory T Cells (CD45RA−/CD45RO+) Dose *Median (Range)*	Timing and n° of Infusion after HSCT *Median (Range), Days*	Immunological Reconstitution *Time, Median (Range) Days*	Clinical Outcome and Side Effects after Memory T Cells Infusion	Reference
N = 17, hematological malignancies, 7.7 (0.6–20.3) years	Haploidentical CD34+ selected	17.8 (3.6–67.5) × 10^6^/kg	121.8 (23.6–528.4) × 10^6^/kg	1 infusion at day +1	Neutrophil engraftment: 10 (9–13) days Platelet engraftment: 17 (10–102) days Full donor chimerism: median day +11	aGVHD: none cGVHD: 35.3% (6 of 17) Viral reactivation: none Relapse: 11.7% (2 of 17) TRM: 11.8% (2 of 17) OS: 76.5%	[28]
N = 8, relapsed or refractory solid tumors, 10 (9–15.2) years	Haploidentical CD34+ selected	10.96 (2.44–34.60) × 10^6^/kg	99.21 (8.56–241.53) × 10^6^/kg	1 infusion at day +1	Neutrophil engraftment: 12 (10–14) day Platelet engraftment: Not defined Full donor chimerism: 13 (10–19) years	aGVHD/cGVHD: none Viral reactivation: none TRM: 12.5% (1 of 8)	[29]
N = 53, N = 36 hematological malignant, N = 17 hematological non-malignant, 9.4 (1–21) years	TCR alpha/beta and CD19 depleted grafts (N = 25, haploidentical; N = 28, MUD).	8.6 (3.2–23) × 10^6^/kg, Residual: TCR αβ+ 12.8 × 10^3^/kg (0.4–331)	25 × 10^3^/kg (haploidentical) 100 × 10^3^/kg cells (MUD)	3 infusions of escalating doses (monthly): 25, 50 and 100 × 10^3^/kg (haploidentical) 100, 200 and 300 × 10^3^/kg (MUD)	Not defined	De novo aGVHD: 2% (1 of 43) Reactivation of preexisting aGVHD: 50% (5 of 10) Viral reactivation: none TRM: 6% OS malignant: 80% OS non-malignant 88% NRM: 5.6% (3 of 53)	[31]
N = 50 high-risk hematological malignancies, 8.1 (0.6–20.8) years	Haploidentical CD34+ selected	16.7 (2.95–67.55) × 10^6^/kg	75.85 (16.1–528.6) × 10^6^/kg	1 infusion at day +1	Neutrophil engraftment: 11 (9–13) days Platelet engraftment: 17 (10–84) days Full donor chimerism: by day +30	aGVHD II-IV grade: 32.1 ± 6.7% aGVHD III-IV grade: 28.1 ± 6.4% Viral reactivation: none Relapse: 18% (9 of 50) OS: 65.8% EFS: 78.9%	[32]
N = 16 hematological malignancies, 54 (30–68) years	HLA-identical family donor 8/8 HLA-MUD	Not defined	1 × 10^6^/kg, 5 × 10^6^/kg, 1 × 10^7^/kg, 5 × 10^7^/kg, 1 × 10^8^/kg.	Dose escalation	Not defined	Relapse 43.7% (7 of 16); death (3 of 7) aGVHD: 6.25% (1 of 16, at dose 5 × 10^6^/kg) cGVHD: 6.25%1 (of 16, at dose 5 × 10^6^/kg) Viral reactivation: none	[33]
N = 19 hematological malignancies, 62 (24–72) years	Haploidentical CD34+	Not defined	5 × 10^5^/kg, 1 × 10^6^/kg, 5 × 10^6^/kg,	Dose escalation (4–6 weeks apart) from day +55 (46–63);	Neutrophil engraftment: 23 (17–34) days Platelet engraftment: 30 (13–75) days Full donor chimerism: 34 (29–49) years	aGVHD II grade: 5% (1 of 19) cGVHD: 10.5% (2 of 19) CMV viremia: 28% (5 of 19) after CD45RA-depleted DLI OS: 79% PFS: 75%	[34]
N = 76 hematological malignancies, 8.6 (0.5–18) years	TCR alpha/beta and CD19 depleted grafts	9 (4–14) × 10^6^/kg, Residual: TCR αβ+ 28 × 10^3^/kg (0.9–361)	25 × 10^3^/kg 50 × 10^3^/kg	1 infusion at day 0; Additional infusion at days: +30, +60, +90, +120	Neutrophil engraftment: 11 (7–24) days Platelet engraftment: 13 (10–33) days Full donor chimerism: by day +30	aGVHD II-IV grade: 14% (11 of 76) aGVHD II-IV grade: 5.3% (4 of 76) CMV viremia: 45% (35 of 76) NRM: 2% Relapse: 25% EFS: 71% OS: 80%,	[35]

## 3. Management of Post-Transplant Viral Infections with T Lymphocytes

Despite the advances in allo-HSCT, the procedure is not without risk, since chronic and refractory viral infections remain a leading cause of morbidity and mortality in the post-transplant period, which is characterized by a prolonged immunodeficient status. It reflects the inability of the depressed host immune system to limit pathogen replication and dissemination [36]. The post-transplant cyclophosphamide protocol is a widely used T-replete strategy and reduces the risk of GVHD by preferentially targeting highly proliferating cells, depleting the alloreactive T cells, including memory T cells that are otherwise pathogen-specific. Similarly, the TCRαβ-depletion and the CD34+-selection results in a loss of antigen-specific memory T cell precursors. Most patients experience post-transplant infections within the first 6 months after HSCT [37,38]. Transplant recipients have an increased susceptibility to viral infections or reactivation caused by most common viruses, such as cytomegalovirus (CMV), Epstein–Barr virus (EBV), BK virus (BKV), adenovirus (AdV), and human herpesvirus 6 (HHV-6). Even GVHD significantly contributes to infectious morbidity and mortality [39]. Since 2017, efforts led by the Board of the Pediatric Diseases Working Party (PDWP) of the European Society for Blood and Marrow Transplantation (EBMT) have been ongoing to develop minimum recommendations tailored specifically to children and young adults undergoing HSCT, emphasizing the importance of pharmacological prophylaxis against viral infections. However, conventional prophylactic and pre-emptive antiviral medications may not always be effective, limited by toxicity, ineffectiveness, the development of drug resistance, and failure to provide long-term protection and immunologic memory. Therefore, strategies to enhance T cell recovery and strengthen pathogen-specific immunity offer promising avenues to complement or replace drug-based approaches [28,32,40,41].

### 3.1. Role of Memory T-Lymphocytes CD45RA−/CD45RO+ to Protect against Viral Infections

Post-transplant infusion of memory T cells (CD45RA−/CD45RO+) can enhance immunological protection without increasing the risk of acute GVHD disease primarily induced by naïve T cells (CD45RA+/CD45RO−) [27]. CD45RA-depleted leukapheresis products, obtained through a clinical manufacturing platform, contain CD4+ and CD8+ effectors and central memory T cells, which sustained IFN-γ secretion against certain viruses previously encountered by the donor [42]. This rapid recall response is faster and more efficient than the primary response mediated by naive T cells. Specifically, memory CD8+ T cells, a subset of cytotoxic T cells, can directly kill infected cells and aim to reduce viral replication and spread. On the other hand, memory CD4+ T cells play a role in activating and guiding other immune cells, such as B cells and cytotoxic T cells. This assists in enhancing the overall immune response to the virus and helps humoral immunity, promoting the generation of antibodies by stimulating B cells to proliferate and differentiate into memory B cells and plasma B cells [43].

Some published studies further emphasize the potential protective effect of adoptive transfer of CD45RA-depleted T cells in allo-HSCT protocols, as summarized in Table 1. For example, in a randomized controlled study of TCRαβ-depleted allogeneic HSCT in high-risk childhood leukemia, post-transplant CD45RA-depleted DLIs were associated with improved recovery of CMV-specific T cells in CMV-seropositive recipients, although the incidence of CMV viremia was not different [35]. Furthermore, a study comparing CD45RA-depleted haplo-HSCT with CD3-depleted haplo-HSCT found better T cell reconstitution and reduced CMV and adenovirus viremia for CD45RA-depleted haplo-HSCT [41].

Intriguingly, pathogen-specific memory T cells could engraft and persist for at least 1 month, explaining the lower incidence of viral infections observed in patients treated with the CD45RA-depleted infusions. However, memory cell infusion can be repeated without complications in persistent infections [31]. Furthermore, the literature data on T cell receptor sequencing analyses showed that CD45RA-depleted DLI does not increase clonal diversity but leads to prominent expansion of a selected number of infused memory clones, suggesting recruitment of these cells in the immune response [44].

### 3.2. Virus-Specific T Lymphocytes vs. T-Memory Lymphocytes CD45RA-/CD45RO+ to Control Symptomatic Viremia and Aspergillosis

To counter ongoing viral infections, clinical-grade virus-specific T cells (VSTs) infusion represents a recently developed approach to control symptomatic viremia [45]. While traditional in vitro expansion methods with cytokines and feeder cells have proven effective in producing functional virus-specific T cells, the CliniMACS^®^ Prodigy system with cytokine capture system (CCS) labeling offers significant improvements in efficiency, safety, and adaptability, positioning it as a superior approach in the evolving landscape of cellular immunotherapy [46]. The CliniMACS^®^ Prodigy manufacturing platform enables fully automated and reproducible separation of viable antigen-specific CD4+ and CD8+ T cells in approximately twelve hours. VSTs can be enriched based on their secretion of IFN-γ after restimulation with single or multiple peptide pools specific to viruses, such as human CMV [47], EBV [48], AdV [49], and BKV [50], or multiple viruses [51]. Commercial peptides are also available to produce *Aspergillus*-specific VSTs [52]. Moreover, the system’s automation capabilities enable more consistent and scalable production, making it particularly advantageous for clinical applications requiring large quantities of high-quality virus-specific T cells. Studies employing VSTs (mono-specific or multi-virus-specific) demonstrate their ability to treat recurrent and refractory viral infections, as summarized in Table 2. Additionally, VSTs can be generated from a stem cell donor or a healthy third-party donor, demonstrating a minimal, if effective, toxicity profile [53]. Third-party donors offer a great opportunity to use the off-the-shelf product for many post-HSCT infections in recipients whose donor lacks virus-specific cellular immune memory [54]. Third-party donors have been shown to have a safe clinical effect and are associated with long-term viral control. However, a higher HLA-match has been associated with better systemic survival of the transferred cells [55,56].

Nevertheless, the enrichment of clinical-grade VSTs from healthy seropositive donors requires aseptic conditions under GMP guidelines and advanced therapy medicinal product (ATMP) regulatory requirements. Extensive manipulation has limitations related to high costs and medium-to-long production times compared to CD45RA-lymphocytes. Memory T lymphocytes offer a viable option for those lacking these requirements, with clinical evidence supporting their effectiveness against refractory viremia. Sanz et colleagues. presented preliminary data on six immunocompromised patients, four with severe infectious diseases and two with EBV lymphoproliferative disease. All patients safely received multiple infusions of familial lymphocytes as adoptive cell therapy, containing memory T cells specific for CMVs, EBV, BKV, and *Aspergillus.* CD45RA-depletion provided many donor memory T cells to the recipients and was associated with enhanced early T cell recovery and protection against viremia [57].

**Table 2 biomedicines-12-01853-t002:** Clinical results of virus-specific T lymphocytes to control post-HSCT viral infections. VST responders are patients who show viral clearance or reduction after receiving VSTs. The aGVHD status is defined as an adverse event following VST infusion effects, including the exacerbation of pre-existing GVHD or other adverse immune reactions immediately after VST infusion. Response to VST treatment is defined only for therapeutic infusions as a decrease of viral copy number by at least 1-log from baseline or clinical improvement or laboratory parameters; partial response is defined as a reduction in viral load greater than 50% and/or improvement in symptomatic disease. aGVHD: acute graft-versus-host disease; ADV: adenovirus; allo-HSCT: allogeneic hematopoietic stem cell transplantation; BKV: BK virus; cGVHD: chronic graft-versus-host disease; CMV: cytomegalovirus; EBV: Epstein–Barr virus; PTLD: post-transplant lymphoproliferative disease; VST: virus-specific T lymphocyte.

Patients *Median (Range), Years*	VST Specificity	VST Doses *Median*	Time/Type of VST Infusion after HSCT *Median (Range), Days*	VST Clinical Response *Only for Therapeutic Infusion:* *R: Responder* *PR: Partial Responder*	Clinical Outcome and Side Effects after VST Infusion	Reference
N = 23, age not specified	CMV	0.1–1 × 10^5^ cell dose/kg	Pre-emptive/prophylactic infusion: 4–130 (median 36) days	Not defined	aGVHD I grade: 13% (3 of 23) No infusion-related toxicities No infection-related mortality	[58]
N = 18, age not specified	CMV	1 × 10^4^ cell dose/kg	N = 11, pre-emptive infusion: 28 (25.5–31) days N = 7, prophylactic infusion: 41 (41–41.5) days	Not defined	aGVHD I-III grade: 16.6% (3 of 18) cGVHD I-III grade: 16.6% (3 of 18) No infusion-related toxicities No infection-related mortality	[47]
N = 18, 18 (7.5–40.7) years	CMV	21.3 × 10^3^ cell dose/kg	Therapeutic infusion: 106.5 (65.5–153.5) days	R: 83.4% (15 of 18)	aGVHD I grade: 16.6% (3 of 18) No infusion-related toxicities Infection-related mortality: 5.5% (1 of 18)	[59]
N = 9, 8 (6–10) years	ADV	1.2–50 × 10^3^ cell dose/kg	Therapeutic infusion: 77 (62–97) days	R: 55.5% (5 of 9)	aGVHD I grade: 11.1% (1 of 9) No infusion-related toxicities Infection-related mortality: 33% (3 of 9) Relapse-related mortality: 11% (1 of 9)	[49]
N = 5	ADV	1.5 × 10^3^ cell dose/kg	Therapeutic infusion: 28 (13–61) days	R: 90% (4 of 5)	aGVHD I–III: 22.2% (2 of 9, documented before VST infusion and worsened) No infusion-related toxicities Infection-related mortality: 11.1% (1 of 9)	[60]
N = 27, 12.15 (0.96–43.3) years	ADV	1.5 × 10^6^ cell dose/kg	Therapeutic infusion: days not defined	R: 54% (15 of 27) PR: 27% (8 of 27)	aGVHD I grade: 11.1% (1 of 27) No infusion-related toxicities Infection-related mortality: 11.1% (3 of 27)	[61]
N = 6, 30 (24–42) years	EBV	7.3 × 10^4^ cell dose/kg	Therapeutic infusion (positive PTLD): days not defined	R: 50% (3 of 6, PTLD complete remission)	No GVHD No infusion-related toxicities Infection-related mortality: 50% (3 of 6 due to multiorgan failure)	[62]
N = 114, 8.4 (0.5–38) years	EBV	0.6–4 × 10^6^ cell dose/kg	Prophylactic infusion (N = 101)/Therapeutic infusion (positive PTLD, N = 13): days not defined	Not defined	aGVHD I–II grade: 7% (8 of 114) cGVHD I–III grade: 12% (13 of 108) Infusion-related toxicities: localized swelling at sites of responsive disease Infection-related mortality: 2.6% (3 of 114) EBV-PTLD related mortality: 1.7% (2 of 114)	[63]
N = 10, 15 (8.3–23.5) years	EBV	2.5–5 × 10^4^ cell dose/kg	Therapeutic infusion: 131 (97–188) days	R: 70% (7 of 10)	aGVHD I–IV grade: 50% (5 of 10) No infusion-related toxicities No Infection-related mortality: 20% (2 of 10, death related to multi-organ failure and relapse) Infection-related mortality: 40% (4 of 10, EBV-PTLD or others)	[48]
N = 16, Mean (SD): 35.6 (21.6) years	BKV	6 × 10^5^ cell dose/kg	Therapeutic infusion: days not defined	R: 100% (16 of 16)	No GVHD No infusion-related toxicities No Infection-related mortality	[54]
N = 6, 15 (9.8–18.8) years	BKV	1.6–6 × 10^5^ cell dose/kg	Therapeutic infusion: 75 (60.3–101.8) days	R: 100% (6 of 6)	No GVHD No infusion-related toxicities No Infection-related mortality: 33% (2 of 6 primary disease complications)	[64]
N = 59, 47 (16–69) years	BKV	2 × 10^5^ cell dose/kg	Therapeutic infusion: 60 (60.3–101.8) days	R: 69.4% (34 of 49) PR: 12.2% (6 of 49)	aGVHD I–II grade: 3.3% (2 of 59) No infusion-related toxicities No Infection-related mortality: 20% (5 of 59, primary disease progression)	[65]

## 4. Adoptive Infusion of Treg Lymphocytes in the Promotion of Immune Tolerance

Despite an overall improvement in incidence and mortality rates during the last 30 years, GVHD remains a major cause of non-relapse mortality after allo-HSCT. Innovative treatments, such as immunotherapy, are critically needed to reduce GVHD prevalence and severity to improve HSCT patient outcomes, as well as to reduce toxicities associated with long-term drug therapy [66]. GVHD occurs in two forms, acute and chronic, which have distinct clinical features that can sometimes present concomitantly or independently of the time after transplant. According to the new diagnostic criteria, acute GVHD is caused by donor T cells reacting to mismatched host polymorphic histocompatibility antigens, leading to an inflammatory cascade. Chronic GVHD shares some traits with autoimmune diseases and can arise de novo or as an extension of acute GVHD [67,68,69]. Even though they account for just 5% of the total CD4 T cell population, it is well established that Tregs (CD4+CD25+Foxp3+) are essential for controlling both acute and chronic GVHD following allo-HSCT, and several studies support the inverse correlation between circulating Treg levels and GVHD occurrence [70,71,72,73,74]. Tregs are critical for maintaining immune tolerance towards self-antigens through several mechanisms of immune regulation, parallel with the intrathymic deletion of self-reactive T cells during ontogeny [75]. They exert regulatory functions through the secretion of anti-inflammatory cytokines, such as IL-10 and TGF-β, the direct cytolysis of effector T cells, the modulation of dendritic cell function, and the metabolic disruption of allogeneic T cells via the high-affinity IL-2 receptor [76]. However, in acute GVHD, studies observed numerically reduced and lower expression levels of granzyme A, CCR5, and CXCR3 in Tregs, suggesting that Treg-suppressive function and target tissue homing may also be impaired, with diminished migration capacity to the target organs [77]. Even patients with chronic GVHD have numerically reduced or functionally impaired circulating Tregs, but the mechanisms that lead to this deficiency in Tregs after allo-HSCT are less understood [78]. An impaired thymic generation of naive Tregs (CD45RA+) and increased apoptosis sensitivity of proliferating memory/effector Tregs (CD45RA−/CD45RO+) could selectively alter Treg homeostasis and contribute to the inability to maintain adequate numbers of Tregs relative to conventional effector CD4+T cells [79].

Current therapeutic approaches for GVHD include immunosuppressive drugs that broadly target immune cells, but these can lead to increased infection susceptibility and relapse of the underlying disease [66]. Based on successes in preclinical GVHD models, adoptive cell therapy with Tregs is promising to induce allograft tolerance or reduce the use of immunosuppressive drugs to prevent rejection. Moreover, clinical trials report manufacturing success and excellent safety and efficacy profiles (Table 3). Clinical studies have demonstrated that post-HSCT infusion of Tregs supports immune reconstitution, helps to maintain immune system balance, and promotes tolerance. Moreover, Tregs aid in tissue repair by reducing inflammation and suppressing excessive immune activation without increasing the risk of relapse and infection [80,81,82,83].

Modulation of Tregs for therapeutic use has become an important area of investigation in GVHD, in which studies are focused on in vitro expanded polyclonal Treg. However, the CliniMACS^®^ Plus provided by Miltenyi Biotech enables us to obtain clinical-grade products, enriching Tregs or depleting non-Treg cells (CD19, CD8, or CD127) selectively through magnetic microbead labeling. Currently, the approaches most cited are CD25 enrichment with or without prior selective depletion of CD8 and/or CD19 cells [84,85,86]. More recently, clinical-grade protocols applied to CliniMACS^®^ Prodigy and combined with MACSQuant^®^ Tyto^®^ Cell Sorter are becoming notable for Treg isolation, allowing the highest purity of Treg product recovery. This manufacturing uses a clinical-grade CD25 GMP reagent to fluorescently label CD25-positive cells; positive cells are purified with a cell sorter following the CD25 enrichment step. In addition, other fluorescent markers, such as CD4, CD127, or CD45RA, can be included for further labeling and sorting. This approach significantly reduces processing time while maintaining product yield and purity. In addition, Tregs can be expanded in the CliniMACS Prodigy^®^ CentriCult Unit using clinical-grade cell culture medium, rapamycin, IL-2, and αCD3/αCD28 beads for 13–14 days, classifying the cell product as ATMP. Successful removal of the expansion beads and the final formulation into the automated procedure results in a ready-to-use product [84].

**Table 3 biomedicines-12-01853-t003:** Clinical results using adoptive cell therapy with allogeneic Tregs to induce allograft tolerance and prevent relapse. Side effects include the exacerbation of pre-existing GVHD or other adverse immune reactions immediately after Treg infusions. aGVHD: acute graft-versus-host disease; cGVHD: chronic graft-versus-host disease; NRM: non-relapse mortality; TRM: transplant-related mortality.

Patients *Median (Range), Year*	T Reg Manufacturing	Treg Cell Dose *Median (Range)*	Treg Clinical Indications	Timing of Infusion after HSCT	Clinical Outcome and Side Effects after Treg Infusion	Reference
N = 19, 39 (20–64) years	Expanded	1–3 × 10^5^ cell dose/kg	aGVHD prophylaxis/ immunological reconstitution	1 infusion at day 0	aGVHD II–IV grade: 26.3% (5 of 19) Relapse: 10.5% (2 of 19) TRM: 31.5% (6 of 19) No infusion-related toxicities	[87]
N = 5, 49 (41–54) years	Expanded	0.5–4.5 × 10^6^ cell dose/kg	Refractory cGVHD/relapse prevention	1 infusion after 35 (35–40) months post-HSCT; double infusion in N = 1 patients.	GVHD improvement: 40% (2 of 5) GVHD stable: 60% (3 of 5) No infusion-related toxicities NRM: 40% (2 of 5) No relapse	[86]
N = 24, 45 (27–61) years	Expanded	2.6 × 10^6^ cell dose/kg	aGVHD prophylaxis/immunological reconstitution	1 infusion at day 0	aGVHD II–IV grade: 33% (8 of 24) cGVHD: 25% (6 of 24) Relapse: 20.8% (5 of 24) No infusion-related toxicities NRM: 4.2% (1 of 24)	[88]
N = 28, 41 (21–60) years	Freshly isolated	2–4 × 10^6^ cell dose/kg	aGVHD prophylaxis/immunological reconstitution	1 infusion at day −4	aGVHD > II grade: 15% (6 of 41) Relapse: 5% (2 of 28) No infusion-related toxicities	[89]
N = 24, 53 (43–56) years	Freshly isolated	1–3 × 10^6^ cell dose/kg	aGVHD prophylaxis/immunological reconstitution	1 infusion at day 0	No aGVHD/cGVHD No infusion-related toxicities No relapse	[90]
N = 41, 40 (18–65) years	Freshly isolated	2.5 × 10^6^ cell dose/kg	Relapse prevention	1 infusion at day 0	aGVHD > II grade: 15% (6 of 41) Relapse: 4.9% (2 of 41) No infusion-related toxicities TRM: 7.5% (3 of 41) NRM: 41.8% (18 of 41)	[91]

## 5. Role of T Lymphocytes in *Graft-Versus-Leukemia* and Relapse

In addition to infection and GVHD, relapse after allo-HSCT remains another therapeutic challenge and the foremost cause of mortality after transplant. The cumulative incidence of relapse for acute leukemia can be as high as 40–50%, with only 10–15% long-term survival in patients experiencing leukemia recurrence after transplant [92]. Combining deeper genetic and molecular characterization, prophylactic strategies, and preventive interventions could already increase survival in high-risk hematological malignancies.

In the pre-DLI era, patients who experienced transplant failure or disease recurrence only had the option of undergoing a second transplant. Adoptive immunotherapy with DLI has been recognized as a treatment option for relapses following allo-HSCT since the early 1990s. The first report demonstrating DLI-induced disease remission after HSCT was published by Kolb and colleagues in 1990 [93]. Over the years, immunotherapy through non-manipulated DLI protocols has been implemented in standard care for patients with mixed chimerism and molecular relapse (prophylactic and pre-emptive DLI) and as maintenance therapy [94,95,96,97]. Data from clinical practice suggest that the best response rates are observed in patients with chronic myeloid leukemia, followed by patients with lymphoma, multiple myeloma, and acute leukemia, respectively. Compared to the others, responses in patients with chronic myeloid leukemia are durable [98]. As mentioned, the choice of HSCT donor does not always fall on HLA-identical available donors but rather on partially compatible donors. Concerning DLI, it is commonly feared that haplo-DLI may lead to a higher risk of GVHD, given the higher degree of HLA disparity between donor and recipient. However, greater HLA disparity may also be helpful in promoting a stronger GVL effect for high-risk acute leukemia [99].

Patients with DLI-responsive relapse usually respond within 2–3 months. Multiple infusions with increasing doses of DLI can be administered until complete remission is achieved or the patient develops clinically significant GVHD [98]. The protocol for DLI after HSCT must be customized for each patient, considering specific clinical features, HLA compatibility, and responses to immunotherapy. Published studies have used wide-ranging escalating cell doses for pre-emptive or prophylactic DLI starting from 1 × 10^5^ CD3+ T cells/kg up to 1 × 10^8^ CD3+ T cells/kg in relapse. The time between doses may vary from 4 to 8 weeks to monitor responses and side effects [94,100]. However, as DLIs contain considerable numbers of alloreactive CD3+/αβ-TCR+ T cells, the clinical-grade manipulation of T lymphocytes with the CliniMACS^®^ systems offers a promising solution to an increase in the number of cells/kg infused, maintaining the balance between GVL effect and reduced GVHD complications. However, it requires advanced technology and significant resources, making it expensive and complex to implement compared to conventional DLI. This allows for personalized treatment and improves non-relapse mortality and overall survival. It is well known that protection from relapse is partly due to donor T cell-mediated GVL immune responses. Manipulated DLI with TCRαβ-depletion is an advanced technique to improve the efficacy and safety of T lymphocyte infusions from haploidentical donors [101]. This manipulation is recommended when rapid intervention is required with an improved safety profile, especially in non-malignant hematological disease in pediatrics. Despite the depletion of CD3+/TCRαβ+ T lymphocytes, the presence of CD3+/TCRγδ+ T lymphocytes and other immune cells, such as NK and Treg cells, can maintain a robust anti-tumor effect for the prevention of recurrence [102,103]. TCRγδ+ T lymphocytes and NK cells are cell types of notable interest because, unlike CD3+ αβ-TCR+ alloreactive T cells that recognize minor or MHC mismatched patient antigens and, thus, can induce GVHD, they recognize their target cells in a non-MHC restricted manner [104]. Even when combining Tregs and conventional T DLI, a very low incidence of relapse (4%) and 75% GVHD/chronic relapse-free survival was observed in adult patients with acute leukemia after haplo-HSCT without post-transplant immunosuppression [105]. Moreover, as reported in Table 3, Tregs when co-infused with conventional T cells (unmanipulated CD3+-DLI) favored post-transplant immune reconstitution and prevented lethal GVHD. The authors evaluated the impact of an early infusion of Tregs, followed by conventional T cells, on GVHD prevention and immunologic reconstitution in 28 patients with high-risk hematologic malignancies who underwent haplo-HSCT [89]. Another cell type worthy of attention in post-transplantation adoptive cell therapy are the stem-like memory T cells (T_SCM_), which constitute a sub-population of long-lived human memory T cells with an increased capacity for self-renewal and the ability to develop into central, effector, and effector memory T cells upon TCR engagement. Due to their proliferative capabilities, T_SCM_ could be ideal weapon for cancer immunotherapy, contributing to the GVL effect. Moreover, several issues regarding T_SCM_ cell biology and mechanism remain to be addressed [106,107,108].

## 6. Conclusions

Improved understanding of T cell biology and the role of DLI after allo-HSCT has sparked a growing interest in the development of not only CAR-T immunotherapeutic strategies. Numerous studies have explored the powerful prophylactic and therapeutic potential of T cell subtypes in adult and pediatric patients, highlighting their profound impact on clinical outcomes after allo-HSCT. As research continues to evolve, further exploration of these mechanisms and their clinical applications will be critical to advance immunotherapeutic approaches in allo-HSCT as a treatment to further improve post-transplant outcomes.

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
