# Peer review of "Advancing Allogeneic Hematopoietic Stem Cell Transplantation Outcomes through Immunotherapy: A Comprehensive Review of Optimizing Non-CAR Donor T-Lymphocyte Infusion Strategies"

_biomedicines, 2024, doi:10.3390/biomedicines12081853_

Round 1
Reviewer 1 Report
Comments and Suggestions for Authors
1. The authors did not mention the different sub-types of memory T cells, particularly the stem cell-like memory T cells. Please provide an explanation regarding these cells in the context of HSCT.
2. Please use recent studies. The references are almost old.
3. It seems that the authors aimed to advertise the benefits of cliniMACS device in the context of donor memory T cells in HSCT. Please provide information on other techniques that can be used in this setting.
Author Response
Comment 1: The authors did not mention the different sub-types of memory T cells, particularly the stem cell-like memory T cells. Please provide an explanation regarding these cells in the context of HSCT.
Response 1:
We thank the reviewer for pointing this out. Stem cell-like memory T cells (TSCM) constitute a long-lived human memory T cell sub-population with an enhanced capacity for self-renewal and the ability to develop into central memory, effector memory, and effector T cells upon TCR engagement. Their role could be exploited therapeutically to improve the efficacy of vaccines and adoptive T-cell therapies for cancer and infectious diseases. Thanks to proliferative abilities, TSCM are the ideal weapon for cancer immunotherapy. There are strategies to induce proliferation and tumor suppression to reach the required cell level for adoptive cell therapy, contributing to the graft-versus-leukemia effect, which is crucial for eradicating residual malignant cells post-transplant. As suggested, we revised the manuscript by adding information about TSCM (Lines 351-357) and clarifying the importance of TSCM in post-transplant cell therapy malignancies. Additionally, they are particularly valuable in CAR-T therapy (PMID: 370184159, PMID: 35503659). Although we chose not to discuss CAR-Ts in detail in the manuscript, delving into this aspect may confuse the reader.
Comment 2: Please use recent studies. The references are almost old.
Response 2: We thank the reviewer for the comment. We revised the manuscript accordingly.
Comment 3: It seems that the authors aimed to advertise the benefits of cliniMACS device in the context of donor memory T cells in HSCT. Please provide information on other techniques that can be used in this setting.
Response 3:
We appreciate the reviewer's comments. In addition to the CliniMACS Plus or Prodigy, the Gibco CTS Rotea Counterflow Centrifugation System seems to be a cell processing system that offers flexibility to address processing needs from cell therapy research to process development to commercial manufacturing.
To selectively isolate memory T cells other techniques could be used. Fluorescence-Activated Cell Sorting (FACS) protocols allow for the isolation of memory T cells from the donor graft. Specific surface markers can identify and separate memory T cells from naïve T cells and other cell types. However, common FACS instruments do not allow cells to be infused safely into patients, but only for research use, as FACS is not a closed GMP system suitable for handling for clinical use. To the best of our knowledge, MACSQuant® Tyto® is the only one with a fully closed cartridge that can process by sorting cells for clinically approved therapeutic protocols. Furthermore, similar to the process performed in closed systems, magnetic beads combined with antibodies targeting specific cell surface markers enable the isolation of various cell populations using columns and separators. The technique could selectively increase or decrease specific cell populations, such as memory T cells. It is less complex and more cost-effective than FACS, although it is generally less precise. The authors emphasize the importance of conducting clinical-grade manipulation using closed systems and adhering to Good Manufacturing Practice (GMP) guidelines when infusing manipulated cells into the patient.
Reviewer 2 Report
Comments and Suggestions for Authors
The manuscript provides a detailed and thorough overview of the role of multiple roles of T lymphocytes in HSCT therapy. However, the discussion of some parts is not sufficient. The author should try to study more articles to give solid discussion to support the conclusions. Thus, it can not be recommended acceptance at the current state. The reviewer suggests it would be resubmitted after a minor revision.
Some specific comments and questions are addressed below.
- The immune regulatory microenvironment in vivo after stem cell transplantation is highly complex, involving the coordinated work of numerous immune cells. How do T lymphocytes collaborate with them? Additionally, how is the balance between pro-inflammatory and anti-inflammatory responses achieved?
- Please check the full name and abbreviation in this article. Full name should be used for the appearance at first time.
Comments on the Quality of English Language
Good
Author Response
General comment: The manuscript provides a detailed and thorough overview of the role of multiple roles of T lymphocytes in HSCT therapy. However, the discussion of some parts is not sufficient. The author should try to study more articles to give solid discussion to support the conclusions. Thus, it can not be recommended acceptance at the current state. The reviewer suggests it would be resubmitted after a minor revision.
Response: We thank the reviewer for his/her feedback on our manuscript. We appreciate the reviewer's recognition of our comprehensive overview of T lymphocytes in HSCT therapy. We improved the manuscript.
Comment 1: The immune regulatory microenvironment in vivo after stem cell transplantation is highly complex, involving the coordinated work of numerous immune cells. How do T lymphocytes collaborate with them? Additionally, how is the balance between pro-inflammatory and anti-inflammatory responses achieved?
Response 1: We thank the reviewer for pointing this out. As suggested, we revised the manuscript by adding information about the immune regulatory microenvironment, focusing on the post-transplant microenvironment in lines 53-77.
Comment 2: Please check the full name and abbreviation in this article. Full name should be used for the appearance at first time.
Response 2: We thank the reviewer for his/her comment. We have revised and modified the manuscript according to these suggestions.
Round 2
Reviewer 1 Report
Comments and Suggestions for Authors
N/A